# Heterogeneous Multi-output Gaussian Process Prediction

**Pablo Moreno-Muñoz**[1]     **Antonio Artés-Rodríguez**[1]     **Mauricio A. Álvarez**[2]

[1]Dept. of Signal Theory and Communications, Universidad Carlos III de Madrid, Spain
[2]Dept. of Computer Science, University of Sheffield, UK

`{pmoreno,antonio}@tsc.uc3m.es, mauricio.alvarez@sheffield.ac.uk`

## Abstract

We present a novel extension of multi-output Gaussian processes for handling heterogeneous outputs. We assume that each output has its own likelihood function and use a vector-valued Gaussian process prior to jointly model the parameters in all likelihoods as latent functions. Our multi-output Gaussian process uses a covariance function with a linear model of coregionalisation form. Assuming conditional independence across the underlying latent functions together with an inducing variable framework, we are able to obtain tractable variational bounds amenable to stochastic variational inference. We illustrate the performance of the model on synthetic data and two real datasets: a human behavioral study and a demographic high-dimensional dataset.

## 1 Introduction

Multi-output Gaussian processes (MOGP) generalise the powerful Gaussian process (GP) predictive model to the vector-valued random field setup (Alvarez et al., 2012). It has been experimentally shown that by simultaneously exploiting correlations between multiple outputs and across the input space, it is possible to provide better predictions, particularly in scenarios with missing or noisy data (Bonilla et al., 2008; Dai et al., 2017).

The main focus in the literature for MOGP has been on the definition of a suitable cross-covariance function between the multiple outputs that allows for the treatment of outputs as a single GP with a properly defined covariance function (Alvarez et al., 2012). The two classical alternatives to define such cross-covariance functions are the linear model of coregionalisation (LMC) (Journel and Huijbregts, 1978) and process convolutions (Higdon, 2002). In the former case, each output corresponds to a weighted sum of shared latent random functions. In the latter, each output is modelled as the convolution integral between a smoothing kernel and a latent random function common to all outputs. In both cases, the unknown latent functions follow Gaussian process priors leading to straight-forward expressions to compute the cross-covariance functions among different outputs. More recent alternatives to build valid covariance functions for MOGP include the work by Ulrich et al. (2015) and Parra and Tobar (2017), that build the cross-covariances in the spectral domain.

Regarding the type of outputs that can be modelled, most alternatives focus on multiple-output regression for continuous variables. Traditionally, each output is assumed to follow a Gaussian likelihood where the mean function is given by one of the outputs of the MOGP and the variance in that distribution is treated as an unknown parameter. Bayesian inference is tractable for these models.

In this paper, we are interested in the heterogeneous case for which the outputs are a mix of continuous, categorical, binary or discrete variables with different likelihood functions.

There have been few attempts to extend the MOGP to other types of likelihoods. For example, Skolidis and Sanguinetti (2011) use the outputs of a MOGP for jointly modelling several binary classification problems, each of which uses a probit likelihood. They use an intrinsic coregionalisation model (ICM), a particular case of LMC. Posterior inference is perfomed using expectation-propagation (EP) and variational mean field. Both Chai (2012) and Dezfouli and Bonilla (2015) have also used ICM for modeling a *single categorical variable* with a multinomial logistic likelihood. The outputs of the ICM model are used as replacements for the linear predictors in the softmax function. Chai (2012) derives a particular variational bound for the marginal likelihood and computes Gaussian posterior distributions; and Dezfouli and Bonilla (2015) introduce an scalable inference procedure that uses a mixture of Gaussians to approximate the posterior distribution using automated variational inference (AVI) (Nguyen and Bonilla, 2014a) that requires sampling from univariate Gaussians.

For the single-output GP case, the usual practice for handling non-Gaussian likelihoods has been replacing the parameters or linear predictors of the non-Gaussian likelihood by one or more *independent GP priors*. Since computing posterior distributions becomes intractable, different alternatives have been offered for approximate inference. An example is the Gaussian heteroscedastic regression model with variational inference (Lázaro-Gredilla and Titsias, 2011), Laplace approximation (Vanhatalo et al., 2013); and stochastic variational inference (SVI) (Saul et al., 2016). This last reference uses the same idea for modulating the parameters of a Student-*t* likelihood, a log-logistic distribution, a beta distribution and a Poisson distribution. The generalised Wishart process (Wilson and Ghahramani, 2011) is another example where the entries of the scale matrix of a Wishart distribution are modulated by independent GPs.

Our main contribution in this paper is to provide an extension of multiple-output Gaussian processes for prediction in heterogeneous datasets. The key principle in our model is to use the outputs of a MOGP as the latent functions that modulate the parameters of several likelihood functions, one likelihood function per output. We tackle the model's intractability using variational inference. Furthermore, we use the inducing variable formalism for MOGP introduced by Alvarez and Lawrence (2009) and compute a variational bound suitable for stochastic optimisation as in Hensman et al. (2013). We experimentally provide evidence of the benefits of simultaneously modeling heterogeneous outputs in different applied problems. Our model can be seen as a generalisation of Saul et al. (2016) for multiple correlated output functions of an heterogeneous nature. Our Python implementation follows the spirit of Hadfield et al. (2010), where the user only needs to specify a list of likelihood functions `likelihood_list = [Bernoulli(), Poisson(), HetGaussian()]`, where `HetGaussian` refers to the heteroscedastic Gaussian distribution, and the number of latent parameter functions per likelihood is assigned automatically.

## 2 Heterogeneous Multi-output Gaussian process

Consider a set of output functions $\mathcal{Y} = \{y_d(\mathbf{x})\}_{d=1}^{D}$, with $\mathbf{x} \in \mathbb{R}^p$, that we want to jointly model using Gaussian processes. Traditionally, the literature has considered the case for which each $y_d(\mathbf{x})$ is continuous and Gaussian distributed. In this paper, we are interested in the heterogeneous case for which the outputs in $\mathcal{Y}$ are a mix of continuous, categorical, binary or discrete variables with several different distributions. In particular, we will assume that the distribution over $y_d(\mathbf{x})$ is completely specified by a set of parameters $\boldsymbol{\theta}_d(\mathbf{x}) \in \mathcal{X}^{J_d}$, where we have a generic $\mathcal{X}$ domain for the parameters and $J_d$ is the number of parameters thet define the distribution. Each parameter $\theta_{d,j}(\mathbf{x}) \in \boldsymbol{\theta}_d(\mathbf{x})$ is a non-linear transformation of a Gaussian process prior $f_{d,j}(\mathbf{x})$, this is, $\theta_{d,j}(\mathbf{x}) = g_{d,j}(f_{d,j}(\mathbf{x}))$, where $g_{d,j}(\cdot)$ is a deterministic function that maps the GP output to the appropriate domain for the parameter $\theta_{d,j}$.

To make the notation concrete, let us assume an heterogeneous multiple-output problem for which $D = 3$. Assume that output $y_1(\mathbf{x})$ is binary and that it will be modelled using a Bernoulli distribution. The Bernoulli distribution uses a single parameter (the probability of success), $J_1 = 1$, restricted to values in the range $[0, 1]$. This means that $\boldsymbol{\theta}_1(\mathbf{x}) = \theta_{1,1}(\mathbf{x}) = g_{1,1}(f_{1,1}(\mathbf{x}))$, where $g_{1,1}(\cdot)$ could be modelled using the logistic sigmoid function $\sigma(z) = 1/(1 + \exp(-z))$ that maps $\sigma : \mathbb{R} \to [0, 1]$. Assume now that the second output $y_2(\mathbf{x})$ corresponds to a count variable that can take values $y_2(\mathbf{x}) \in \mathbb{N} \cup \{0\}$. The count variable can be modelled using a Poisson distribution with a single parameter (the rate), $J_2 = 1$, restricted to the positive reals. This means that $\boldsymbol{\theta}_2(\mathbf{x}) = \theta_{2,1}(\mathbf{x}) = g_{2,1}(f_{2,1}(\mathbf{x}))$ where $g_{2,1}(\cdot)$ could be modelled as an exponential function $g_{2,1}(\cdot) = \exp(\cdot)$ to ensure strictly positive values for the parameter. Finally, $y_3(\mathbf{x})$ is a continuous variable with heteroscedastic

noise. It can be modelled using a Gaussian distribution where both the mean and the variance are functions of $\mathbf{x}$. This means that $\boldsymbol{\theta}_3(\mathbf{x}) = [\theta_{3,1}(\mathbf{x}) \quad \theta_{3,2}(\mathbf{x})]^\top = [g_{3,1}(f_{3,1}(\mathbf{x})) \quad g_{3,2}(f_{3,2}(\mathbf{x}))]^\top$, where the first function is used to model the mean of the Gaussian, and the second function is used to model the variance. Therefore, we can assume the $g_{3,1}(\cdot)$ is the identity function and $g_{3,2}(\cdot)$ is a function that ensures that the variance takes strictly positive values, e.g. the exponential function.

Let us define a vector-valued function $\mathbf{y}(\mathbf{x}) = [y_1(\mathbf{x}), y_2(\mathbf{x}), \cdots, y_D(\mathbf{x})]^\top$. We assume that the outputs are conditionally independent given the vector of parameters $\boldsymbol{\theta}(\mathbf{x}) = [\theta_1(\mathbf{x}), \theta_2(\mathbf{x}), \cdots, \theta_D(\mathbf{x})]^\top$, defined by specifying the vector of latent functions $\mathbf{f}(\mathbf{x}) = [f_{1,1}(\mathbf{x}), f_{1,2}(\mathbf{x}), \cdots f_{1,J_1}(\mathbf{x}), f_{2,1}(\mathbf{x}), f_{2,2}(\mathbf{x}), \cdots, f_{D,J_D}(\mathbf{x})]^\top \in \mathbb{R}^{J \times 1}$, where $J = \sum_{d=1}^D J_d$,

$$p(\mathbf{y}(\mathbf{x})|\boldsymbol{\theta}(\mathbf{x})) = p(\mathbf{y}(\mathbf{x})|\mathbf{f}(\mathbf{x})) = \prod_{d=1}^D p(y_d(\mathbf{x})|\boldsymbol{\theta}_d(\mathbf{x})) = \prod_{d=1}^D p(y_d(\mathbf{x})|\widetilde{\mathbf{f}}_d(\mathbf{x})), \qquad (1)$$

where we have defined $\widetilde{\mathbf{f}}_d(\mathbf{x}) = [f_{d,1}(\mathbf{x}), \cdots, f_{d,J_d}(\mathbf{x})]^\top \in \mathbb{R}^{J_d \times 1}$, the set of latent functions that specify the parameters in $\boldsymbol{\theta}_d(\mathbf{x})$. Notice that $J \geq D$. This is, there is not always a one-to-one map from $\mathbf{f}(\mathbf{x})$ to $\mathbf{y}(\mathbf{x})$. Most previous work has assumed that $D = 1$, and that the corresponding elements in $\boldsymbol{\theta}_d(\mathbf{x})$, this is, the latent functions in $\widetilde{\mathbf{f}}_1(\mathbf{x}) = [f_{1,1}(\mathbf{x}), \cdots, f_{1,J_1}(\mathbf{x})]^\top$ are drawn from independent Gaussian processes. Important exceptions are Chai (2012) and Dezfouli and Bonilla (2015), that assumed a categorical variable $y_1(\mathbf{x})$, where the elements in $\widetilde{\mathbf{f}}_1(\mathbf{x})$ were drawn from an intrinsic coregionalisation model. In what follows, we generalise these models for $D > 1$ and potentially heterogeneuos outputs $y_d(\mathbf{x})$. We will use the word "output" to refer to the elements $y_d(\mathbf{x})$ and "latent parameter function" (LPF) or "parameter function" (PF) to refer to $f_{d,j}(\mathbf{x})$.

## 2.1  A multi-parameter GP prior

Our main departure from previous work is in modeling of $\mathbf{f}(\mathbf{x})$ using a multi-parameter Gaussian process that allows correlations for the parameter functions $f_{d,j}(\mathbf{x})$. We will use a linear model of coregionalisation type of covariance function for expressing correlations between functions $f_{d,j}(\mathbf{x})$, and $f_{d',j'}(\mathbf{x}')$. The particular construction is as follows. Consider an additional set of independent latent functions $\mathcal{U} = \{u_q(\mathbf{x})\}_{q=1}^Q$ that will be linearly combined to produce $J$ LPFs $\{f_{d,j}(\mathbf{x})\}_{j=1,d=1}^{J_d,D}$. Each latent function $u_q(\mathbf{x})$ is assummed to be drawn from an independent GP prior such that $u_q(\cdot) \sim \mathcal{GP}(0, k_q(\cdot, \cdot))$, where $k_q$ can be any valid covariance function, and the zero mean is assumed for simplicity. Each latent parameter $f_{d,j}(\mathbf{x})$ is then given as

$$f_{d,j}(\mathbf{x}) = \sum_{q=1}^Q \sum_{i=1}^{R_q} a_{d,j,q}^i u_q^i(\mathbf{x}), \qquad (2)$$

where $u_q^i(\mathbf{x})$ are IID samples from $u_q(\cdot) \sim \mathcal{GP}(0, k_q(\cdot, \cdot))$ and $a_{d,j,q}^i \in \mathbb{R}$. The mean function for $f_{d,j}(\mathbf{x})$ is zero and the cross-covariance function $k_{f_{d,j}f_{d',j'}}(\mathbf{x}, \mathbf{x}') = \text{cov}[f_{d,j}(\mathbf{x}), f_{d',j'}(\mathbf{x}')]$ is equal to $\sum_{q=1}^Q b_{(d,j),(d',j')}^q k_q(\mathbf{x}, \mathbf{x}')$, where $b_{(d,j),(d',j')}^q = \sum_{i=1}^{R_q} a_{d,j,q}^i a_{d',j',q}^i$. Let us define $\mathbf{X} = \{\mathbf{x}_n\}_{n=1}^N \in \mathbb{R}^{N \times p}$ as a set of common input vectors for all outputs $y_d(\mathbf{x})$. Although, the presentation could be extended for the case of a different set of inputs per output. Let us also define $\mathbf{f}_{d,j} = [f_{d,j}(\mathbf{x}_1), \cdots, f_{d,j}(\mathbf{x}_N)]^\top \in \mathbb{R}^{N \times 1}$; $\widetilde{\mathbf{f}}_d = [\mathbf{f}_{d,1}^\top \cdots \mathbf{f}_{d,J_d}^\top]^\top \in \mathbb{R}^{J_d N \times 1}$, and $\mathbf{f} = [\widetilde{\mathbf{f}}_1^\top \cdots \widetilde{\mathbf{f}}_D^\top]^\top \in \mathbb{R}^{JN \times 1}$. The generative model for the heterogeneous MOGP is as follows. We sample $\mathbf{f} \sim \mathcal{N}(\mathbf{0}, \mathbf{K})$, where $\mathbf{K}$ is a block-wise matrix with blocks given by $\{\mathbf{K}_{\mathbf{f}_{d,j}\mathbf{f}_{d',j'}}\}_{d=1,d'=1,j=1,j'=1}^{D,D,J_d,J_{d'}}$. In turn, the elements in $\mathbf{K}_{\mathbf{f}_{d,j}\mathbf{f}_{d',j'}}$ are given by $k_{f_{d,j}f_{d',j'}}(\mathbf{x}_n, \mathbf{x}_m)$, with $\mathbf{x}_n, \mathbf{x}_m \in \mathbf{X}$. For the particular case of equal inputs $\mathbf{X}$ for all LPF, $\mathbf{K}$ can also be expressed as the sum of Kronecker products $\mathbf{K} = \sum_{q=1}^Q \mathbf{A}_q \mathbf{A}_q^\top \otimes \mathbf{K}_q = \sum_{q=1}^Q \mathbf{B}_q \otimes \mathbf{K}_q$, where $\mathbf{A}_q \in \mathbb{R}^{J \times R_q}$ has entries $\{a_{d,j,q}^i\}_{d=1,j=1,i=1}^{D,J_d,R_q}$ and $\mathbf{B}_q$ has entries $\{b_{(d,j),(d',j')}^q\}_{d=1,d'=1,j=1,j'=1}^{D,D,J_d,J_{d'}}$. The matrix $\mathbf{K}_q \in \mathbb{R}^{N \times N}$ has entries given by $k_q(\mathbf{x}_n, \mathbf{x}_m)$ for $\mathbf{x}_n, \mathbf{x}_m \in \mathbf{X}$. Matrices $\mathbf{B}_q \in \mathbb{R}^{J \times J}$ are known as the *coregionalisation matrices*. Once we obtain the sample for $\mathbf{f}$, we evaluate the vector of parameters $\boldsymbol{\theta} = [\boldsymbol{\theta}_1^\top \cdots \boldsymbol{\theta}_D^\top]^\top$, where $\boldsymbol{\theta}_d = \widetilde{\mathbf{f}}_d$. Having specified $\boldsymbol{\theta}$, we can generate samples for the output vector $\mathbf{y} = [\mathbf{y}_1^\top \cdots \mathbf{y}_D^\top]^\top \in \mathcal{X}^{DN \times 1}$, where the elements in $\mathbf{y}_d$ are obtained by sampling from the conditional distributions

$p(y_d(\mathbf{x})|\boldsymbol{\theta}_d(\mathbf{x}))$. To keep the notation uncluttered, we will assume from now that $R_q = 1$, meaning that $\mathbf{A}_q = \mathbf{a}_q \in \mathbb{R}^{J \times 1}$, and the corregionalisation matrices are rank-one. In the literature such model is known as the *semiparametric latent factor model* (Teh et al., 2005).

## 2.2 Scalable variational inference

Given an heterogeneous dataset $\mathcal{D} = \{\mathbf{X}, \mathbf{y}\}$, we would like to compute the posterior distribution for $p(\mathbf{f}|\mathcal{D})$, which is intractable in our model. In what follows, we use similar ideas to Alvarez and Lawrence (2009); Álvarez et al. (2010) that introduce the inducing variable formalism for computational efficiency in MOGP. However, instead of marginalising the latent functions $\mathcal{U}$ to obtain a variational lower bound, we keep their presence in a way that allows us to apply stochastic variational inference as in Hensman et al. (2013); Saul et al. (2016).

### 2.2.1 Inducing variables for MOGP

A key idea to reduce computational complexity in Gaussian process models is to introduce *auxiliary variables* or *inducing variables*. These variables have been used already in the context of MOGP (Alvarez and Lawrence, 2009; Álvarez et al., 2010) . A subtle difference from the single output case is that the inducing variables are not taken from the same latent process, say $f_1(\mathbf{x})$, but from the latent processes $\mathcal{U}$ used also to build the model for multiple outputs. We will follow the same formalism here. We start by defining the set of $M$ inducing variables per latent function $u_q(\mathbf{x})$ as $\mathbf{u}_q = [u_q(\mathbf{z}_1), \cdots, u_q(\mathbf{z}_M)]^\top$, evaluated at a set of *inducing inputs* $\mathbf{Z} = \{\mathbf{z}_m\}_{m=1}^M \in \mathbb{R}^{M \times p}$. We also define $\mathbf{u} = [\mathbf{u}_1^\top, \cdots, \mathbf{u}_Q^\top]^\top \in \mathbb{R}^{QM \times 1}$. For simplicity in the exposition, we have assumed that all the inducing variables, for all $q$, have been evaluated at the same set of inputs $\mathbf{Z}$. Instead of marginalising $\{u_q(\mathbf{x})\}_q^Q$ from the model in (2), we explicitly use the joint Gaussian prior $p(\mathbf{f}, \mathbf{u}) = p(\mathbf{f}|\mathbf{u})p(\mathbf{u})$. Due to the assumed independence in the latent functions $u_q(\mathbf{x})$, the distribution $p(\mathbf{u})$ factorises as $p(\mathbf{u}) = \prod_{q=1}^Q p(\mathbf{u}_q)$, with $\mathbf{u}_q \sim \mathcal{N}(\mathbf{0}, \mathbf{K}_q)$, where $\mathbf{K}_q \in \mathbb{R}^{M \times M}$ has entries $k_q(\mathbf{z}_i, \mathbf{z}_j)$ with $\mathbf{z}_i, \mathbf{z}_j \in \mathbf{Z}$. Notice that the dimensions of $\mathbf{K}_q$ are different to the dimensions of $\mathbf{K}_q$ in section 2.1. The LPFs $\mathbf{f}_{d,j}$ are conditionally independent given $\mathbf{u}$, so we can write the conditional distribution $p(\mathbf{f}|\mathbf{u})$ as

$$p(\mathbf{f}|\mathbf{u}) = \prod_{d=1}^D \prod_{j=1}^{J_d} p(\mathbf{f}_{d,j}|\mathbf{u}) = \prod_{d=1}^D \prod_{j=1}^{J_d} \mathcal{N}\Big(\mathbf{f}_{d,j}|\mathbf{K}_{\mathbf{f}_{d,j}\mathbf{u}}\mathbf{K}_{\mathbf{uu}}^{-1}\mathbf{u}, \mathbf{K}_{\mathbf{f}_{d,j}\mathbf{f}_{d,j}} - \mathbf{K}_{\mathbf{f}_{d,j}\mathbf{u}}\mathbf{K}_{\mathbf{uu}}^{-1}\mathbf{K}_{\mathbf{f}_{d,j}\mathbf{u}}^\top\Big),$$

where $\mathbf{K}_{\mathbf{uu}} \in \mathbb{R}^{QM \times QM}$ is a block-diagonal matrix with blocks given by $\mathbf{K}_q$ and $\mathbf{K}_{\mathbf{f}_{d,j}\mathbf{u}} \in \mathbb{R}^{N \times QM}$ is the cross-covariance matrix computed from the cross-covariances between $f_{d,j}(\mathbf{x})$ and $u_q(\mathbf{z})$. The expression for this cross-covariance function can be obtained from (2) leading to $k_{f_{d,j}u_q}(\mathbf{x}, \mathbf{z}) = a_{d,j,q}k_q(\mathbf{x}, \mathbf{z})$. This form for the cross-covariance between the LPF $f_{d,j}(\mathbf{x})$ and $u_q(\mathbf{z})$ is a key difference between the inducing variable methods for the single-output GP case and the MOGP case.

### 2.2.2 Variational Bounds

Exact posterior inference is intractable in our model due to the presence of an arbitrary number of non-Gaussian likelihoods. We use variational inference to compute a lower bound $\mathcal{L}$ for the marginal log-likelihood $\log p(\mathbf{y})$, and for approximating the posterior distribution $p(\mathbf{f}, \mathbf{u}|\mathcal{D})$. Following Álvarez et al. (2010), the posterior of the LPFs $\mathbf{f}$ and the latent functions $\mathbf{u}$ can be approximated as

$$p(\mathbf{f}, \mathbf{u}|\mathbf{y}, \mathbf{X}) \approx q(\mathbf{f}, \mathbf{u}) = p(\mathbf{f}|\mathbf{u})q(\mathbf{u}) = \prod_{d=1}^D \prod_{j=1}^{J_d} p(\mathbf{f}_{d,j}|\mathbf{u}) \prod_{q=1}^Q q(\mathbf{u}_q),$$

where $q(\mathbf{u}_q) = \mathcal{N}(\mathbf{u}_q|\boldsymbol{\mu}_{\mathbf{u}_q}, \mathbf{S}_{\mathbf{u}_q})$ are Gaussian variational distributions whose parameters $\{\boldsymbol{\mu}_{\mathbf{u}_q}, \mathbf{S}_{\mathbf{u}_q}\}_{q=1}^Q$ must be optimised. Building on previous work by Saul et al. (2016); Hensman et al. (2015), we derive a lower bound that accepts any log-likelihood function that can be modulated by the LPFs $\mathbf{f}$. The lower bound $\mathcal{L}$ for $\log p(\mathbf{y})$ is obtained as follows

$$\log p(\mathbf{y}) = \log \int p(\mathbf{y}|\mathbf{f})p(\mathbf{f}|\mathbf{u})p(\mathbf{u})d\mathbf{f}d\mathbf{u} \geq \int q(\mathbf{f}, \mathbf{u}) \log \frac{p(\mathbf{y}|\mathbf{f})p(\mathbf{f}|\mathbf{u})p(\mathbf{u})}{q(\mathbf{f}, \mathbf{u})} d\mathbf{f}d\mathbf{u} = \mathcal{L}.$$

We can further simplify $\mathcal{L}$ to obtain

$$\mathcal{L} = \int \int p(\mathbf{f}|\mathbf{u})q(\mathbf{u}) \log p(\mathbf{y}|\mathbf{f}) d\mathbf{f} d\mathbf{u} - \sum_{q=1}^{Q} \text{KL}\big(q(\mathbf{u}_q)||p(\mathbf{u}_q)\big)$$

$$= \int \int \prod_{d=1}^{D} \prod_{j=1}^{J_d} p(\mathbf{f}_{d,j}|\mathbf{u})q(\mathbf{u}) \log p(\mathbf{y}|\mathbf{f}) d\mathbf{u} d\mathbf{f} - \sum_{q=1}^{Q} \text{KL}\big(q(\mathbf{u}_q)||p(\mathbf{u}_q)\big),$$

where KL is the Kullback-Leibler divergence. Moreover, the approximate marginal posterior for $\mathbf{f}_{d,j}$ is $q(\mathbf{f}_{d,j}) = \int p(\mathbf{f}_{d,j}|\mathbf{u})q(\mathbf{u})d\mathbf{u}$, leading to

$$q(\mathbf{f}_{d,j}) = \mathcal{N}\Big(\mathbf{f}_{d,j}|\mathbf{K}_{\mathbf{f}_{d,j}\mathbf{u}}\mathbf{K}_{\mathbf{uu}}^{-1}\boldsymbol{\mu}_{\mathbf{u}}, \mathbf{K}_{\mathbf{f}_{d,j}\mathbf{f}_{d,j}} + \mathbf{K}_{\mathbf{f}_{d,j}\mathbf{u}}\mathbf{K}_{\mathbf{uu}}^{-1}(\mathbf{S}_{\mathbf{u}} - \mathbf{K}_{\mathbf{uu}})\mathbf{K}_{\mathbf{uu}}^{-1}\mathbf{K}_{\mathbf{f}_{d,j}\mathbf{u}}^{\top}\Big),$$

where $\boldsymbol{\mu}_{\mathbf{u}} = [\boldsymbol{\mu}_{\mathbf{u}_1}^{\top}, \cdots, \boldsymbol{\mu}_{\mathbf{u}_Q}^{\top}]^{\top}$ and $\mathbf{S}_{\mathbf{u}}$ is a block-diagonal matrix with blocks given by $\mathbf{S}_{\mathbf{u}_q}$. The expression for $\log p(\mathbf{y}|\mathbf{f})$ factorises, according to (1): $\log p(\mathbf{y}|\mathbf{f}) = \sum_{d=1}^{D} \log p(\mathbf{y}_d|\widetilde{\mathbf{f}}_d) = \sum_{d=1}^{D} \log p(\mathbf{y}_d|\mathbf{f}_{d,1}, \cdots, \mathbf{f}_{d,J_d})$. Using this expression for $\log p(\mathbf{y}|\mathbf{f})$ leads to the following expression for the bound

$$\sum_{d=1}^{D} \mathbb{E}_{q(\mathbf{f}_{d,1})\cdots q(\mathbf{f}_{d,J_d})}\big[\log p(\mathbf{y}_d|\mathbf{f}_{d,1}, \cdots, \mathbf{f}_{d,J_d})\big] - \sum_{q=1}^{Q} \text{KL}\big(q(\mathbf{u}_q)||p(\mathbf{u}_q)\big).$$

When $D = 1$ in the expression above, we recover the bound obtained in Saul et al. (2016). To maximize this lower bound, we need to find the optimal variational parameters $\{\boldsymbol{\mu}_{\mathbf{u}_q}\}_{q=1}^{Q}$ and $\{\mathbf{S}_{\mathbf{u}_q}\}_{q=1}^{Q}$. We represent each matrix $\mathbf{S}_{\mathbf{u}_q}$ as $\mathbf{S}_{\mathbf{u}_q} = \mathbf{L}_{\mathbf{u}_q}\mathbf{L}_{\mathbf{u}_q}^{\top}$ and, to ensure positive definiteness for $\mathbf{S}_{\mathbf{u}_q}$, we estimate $\mathbf{L}_{\mathbf{u}_q}$ instead of $\mathbf{S}_{\mathbf{u}_q}$. Computation of the posterior distributions over $\mathbf{f}_{d,j}$ can be done analytically. There is still an intractability issue in the variational expectations on the log-likelihood functions. Since we construct these bounds in order to accept any possible data type, we need a general way to solve these integrals. One obvious solution is to apply Monte Carlo methods, however it would be slow both maximising the lower bound and updating variational parameters by sampling thousands of times (for approximating expectations) at each iteration. Instead, we address this problem by using Gaussian-Hermite quadratures as in Hensman et al. (2015); Saul et al. (2016).

**Stochastic Variational Inference.** The conditional expectations in the bound above are also valid across data observations so that we can express the bound as

$$\sum_{d=1}^{D} \sum_{n=1}^{N} \mathbb{E}_{q(f_{d,1}(\mathbf{x}_n))\cdots q(f_{d,J_d}(\mathbf{x}_n))}\big[\log p(y_d(\mathbf{x}_n)|f_{d,1}(\mathbf{x}_n), \cdots, f_{d,J_d}(\mathbf{x}_n))\big] - \sum_{q=1}^{Q} \text{KL}\big(q(\mathbf{u}_q)||p(\mathbf{u}_q)\big).$$

This functional form allows the use of *mini-batches* of smaller sets of training samples, performing the optimization process using noisy estimates of the global objective gradient in a similar fashion to Hoffman et al. (2013); Hensman et al. (2013, 2015); Saul et al. (2016) . This scalable bound makes our multi-ouput model applicable to large heterogenous datasets. Notice that computational complexity is dominated by the inversion of $\mathbf{K}_{\mathbf{uu}}$ with a cost of $\mathcal{O}(QM^3)$ and products like $\mathbf{K}_{\mathbf{fu}}$ with a cost of $\mathcal{O}(JNQM^2)$.

**Hyperparameter learning.** Hyperparameters in our model include $\mathbf{Z}$, $\{\mathbf{B}_q\}_{q=1}^{Q}$, and $\{\gamma_q\}_{q=1}^{Q}$, the hyperparameters associated to the covariance functions $\{k_q(\cdot, \cdot)\}_{q=1}^{Q}$. Since the variational distribution $q(\mathbf{u})$ is sensitive to changes of the hyperparameters, we maximize the variational parameters for $q(\mathbf{u})$, and the hyperparameters using a Variational EM algorithm (Beal, 2003) when employing the full dataset, or the stochastic version when using mini-batches (Hoffman et al., 2013).

## 2.3 Predictive distribution

Consider a set of test inputs $\mathbf{X}_*$. Assuming that $p(\mathbf{u}|\mathbf{y}) \approx q(\mathbf{u})$, the predictive distribution $p(\mathbf{y}_*)$ can be approximated as $p(\mathbf{y}_*|\mathbf{y}) \approx \int p(\mathbf{y}_*|\mathbf{f}_*)q(\mathbf{f}_*)d\mathbf{f}_*$, where $q(\mathbf{f}_*) = \int p(\mathbf{f}_*|\mathbf{u})q(\mathbf{u})d\mathbf{u}$. Computing the expression $q(\mathbf{f}_*) = \prod_{d=1}^{D} \prod_{j=1}^{J_d} q(\mathbf{f}_{d,j,*})$ involves evaluating $\mathbf{K}_{\mathbf{f}_{d,j,*}\mathbf{u}}$ at $\mathbf{X}_*$. As in the case of

the lower bound, the integral above is intractable for the non-Gaussian likelihoods $p(\mathbf{y}_*|\mathbf{f}_*)$. We can once again make use of Monte Carlo integration or quadratures to approximate the integral. Simpler integration problems are obtained if we are only interested in the predictive mean, $\mathbb{E}[\mathbf{y}_*]$, and the predictive variance, $\mathrm{var}[\mathbf{y}_*]$.

## 3  Related Work

The most closely related works to ours are Skolidis and Sanguinetti (2011), Chai (2012), Dezfouli and Bonilla (2015) and Saul et al. (2016). We are different from Skolidis and Sanguinetti (2011) because we allow more general heterogeneous outputs beyond the specific case of several binary classification problems. Our inference method also scales to large datasets. The works by Chai (2012) and Dezfouli and Bonilla (2015) do use a MOGP, but they only handle a single categorical variable. Our inference approach scales when compared to the one in Chai (2012) and it is fundamentally different to the one in Dezfouli and Bonilla (2015) since we do not use AVI. Our model is also different to Saul et al. (2016) since we allow for several dependent outputs, $D > 1$, and our scalable approach is more akin to applying SVI to the inducing variable approach of Álvarez et al. (2010).

More recently, Vanhatalo et al. (2018) used additive multi-output GP models to account for inter-dependencies between counting and binary observations. They use the Laplace approximation for approximating the posterior distribution. Similarly, Pourmohamad and Lee (2016) perform combined regression and binary classification with a multi-output GP learned via sequential Monte Carlo. Nguyen and Bonilla (2014b) also uses the same idea from Álvarez et al. (2010) to provide scalability for multiple-output GP models conditioning the latent parameter functions $f_{d,j}(\mathbf{x})$ on the inducing variables $\mathbf{u}$, but only considers the multivariate regression case.

It is also important to mention that multi-output Gaussian processes have been considered as alternative models for multi-task learning (Alvarez et al., 2012). Multi-task learning also addresses multiple prediction problems together within a single inference framework. Most previous work in this area has focused on problems where all tasks are exclusively regression or classification problems. When tasks are heterogeneous, the common practice is to introduce a regularizer per data type in a global cost function (Zhang et al., 2012; Han et al., 2017). Usually, these cost functions are compounded by additive terms, each one referring to every single task, while the correlation assumption among heterogeneous likelihoods is addressed by mixing regularizers in a global penalty term (Li et al., 2014) or by forcing different tasks to share a common mean (Ngufor et al., 2015). Another natural way of treating both continuous and discrete tasks is to assume that all of them share a common input set that varies its influence on each output. Then, by sharing a jointly sparsity pattern, it is possible to optimize a global cost function with a single regularization parameter on the level of sparsity (Yang et al., 2009). There have also been efforts for modeling heterogeneous data outside the label of multi-task learning including mixed graphical models (Yang et al., 2014), where varied types of data are assumed to be combinations of exponential families, and latent feature models (Valera et al., 2017) with heterogeneous observations being mappings of a set of Gaussian distributed variables.

## 4  Experiments

In this section, we evaluate our model on different heterogeneous scenarios [1]. To demonstrate its performance in terms of multi-output learning, prediction and scalability, we have explored several applications with both synthetic and real data. For all the experiments, we consider an RBF kernel for each covariance function $k_q(\cdot, \cdot)$ and we set $Q = 3$. For standard optimization we used the LBFGS-B algorithm. When SVI was needed, we considered ADADELTA included in the *climin* library, and a *mini-batch* size of 500 samples in every output. All performance metrics are given in terms of the negative log-predictive density (NLPD) calculated from a test subset and applicable to any type of likelihood. Further details about experiments are included in the appendix.

**Missing Gap Prediction:** In our first experiment, we evaluate if our model is able to predict observations in one output using training information from another one. We setup a toy problem which consists of $D = 2$ heterogeneous outputs, where the first function $y_1(\mathbf{x})$ is real and $y_2(\mathbf{x})$ binary. Assuming that heterogeneous outputs do not share a common input set, we observe

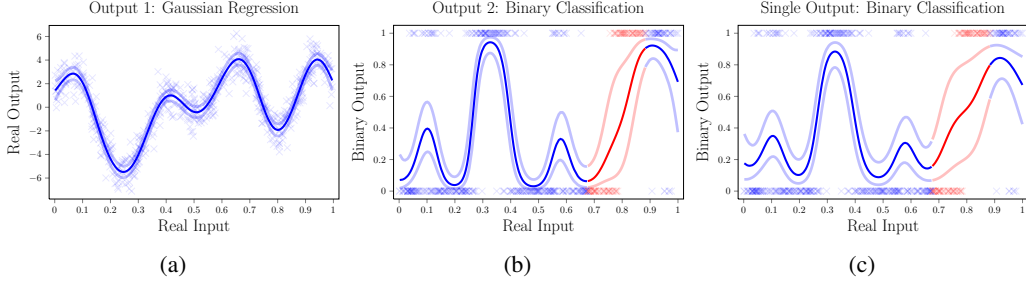

(a)            (b)            (c)

Figure 1: Comparison between multi-output and single-output performance for two heterogeneous sets of observations. a) Fitted function and uncertainty for the first output. It represents the mean function parameter $\mu(\mathbf{x})$ for a Gaussian distribution with $\sigma^2 = 1$. b) Predictive output function for binary inputs. Blue curve is the fitting function for training data and the red one corresponds to predicting from test inputs (true test binary outputs in red too). c) Same output as Figure 1(b) but training an independent Chained GP only in the single binary output (GP binary classification).

$N_1 = 600$ and $N_2 = 500$ samples respectively. All inputs are uniformly distributed in the input range $[0, 1]$, and we generate a *gap* only in the set of binary observations by removing $N_{\text{test}} = 150$ samples in the interval $[0.7, 0.9]$. Using the remaining points from both outputs for training, we fitted our MOGP model. In Figures 1(a,b) we can see how uncertainty in binary test predictions is reduced by learning from the first output. In contrast, Figure 1(c) shows wider variance in the predicted parameter when it is trained independently. For the multi-output case we obtained a NLPD value for test data of $32.5 \pm 0.2 \times 10^{-2}$ while in the single-output case the NLPD was $40.51 \pm 0.08 \times 10^{-2}$.

**Human Behavior Data:** In this experiment, we are interested in modeling human behavior in psychiatric patients. Previous work by Soleimani et al. (2018) already explores the application of scalable MOGP models to healthcare for reliable predictions from multivariate time-series. Our data comes from a medical study that asked patients to download a monitoring *app* (EB2)[2] on their smartphones. The system captures information about mobility, communication metadata and interactions in social media. The work has a particular interest in mental health since shifts or misalignments in the circadian feature of human behavior (24h cycles) can be interpreted as early signs of crisis.

Table 1: Behavior Dataset Test-NLPD ($\times 10^{-2}$)

|  | **Bernoulli** | **Heteroscedastic** | **Bernoulli** | **Global** |
|---|---|---|---|---|
| HetMOGP | **2.24 ± 0.21** | **6.09 ± 0.21** | 5.41 ± 0.05 | **13.74 ± 0.41** |
| ChainedGP | 2.43 ± 0.30 | 7.29 ± 0.12 | **5.19 ± 0.81** | 14.91 ± 1.05 |

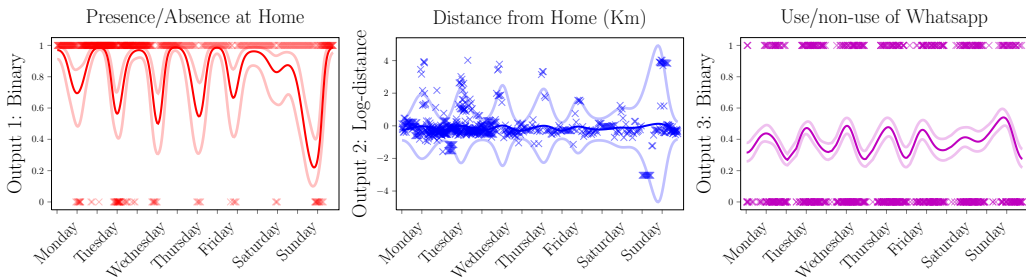

Figure 2: Results for multi-output modeling of human behavior. After training, all output predictions share a common (daily) periodic pattern.

In particular, we obtained a binary indicator variable of presence/absence at home by monitoring *latitude-longitude* and measuring its distance from the patient's home location within a 50m radius range. Then, using the already measured distances, we generated a mobility sequence with all log-distance values. Our last output consists of binary samples representing use/non-use of the

*Whatsapp* application in the smartphone. At each monitoring time instant, we used its differential data consumption to determine use or non-use of the application. We considered an entire week in seconds as the input domain, normalized to the range $[0, 1]$.

In Figure (2), after training on $N = 750$ samples, we find that the circadian feature is mainly contained in the first output. During the learning process, this periodicity is transferred to the other outputs through the latent functions improving the performance of the entire model. Experimentally, we tested that this circadian pattern was not captured in mobility and social data when training outputs independently. In Table 1 we can see prediction metrics for multi-output and independent prediction.

**London House Price Data:** Based on the large scale experiments in Hensman et al. (2013), we obtained the complete register of properties sold in the Greater London County during 2017 (`https://www.gov.uk/government/collections/price-paid-data`). We preprocessed it to translate all property addresses to *latitude-longitude* points. For each spatial input, we considered two observations, one binary and one real. The first one indicates if the property is or is not a flat (zero would mean detached, semi-detached, terraced, etc.. ), and the second one the sale price of houses. Our goal is to predict features of houses given a certain location in the London area. We used a training set of $N = 20,000$ samples, $1,000$ for test predictions and $M = 100$ inducing points.

Table 2: London Dataset Test-NLPD ($\times 10^{-2}$)

|  | **Bernoulli** | **Heteroscedastic** | **Global** |
|---|---|---|---|
| HetMOGP | $\mathbf{6.38 \pm 0.46}$ | $\mathbf{10.05 \pm 0.64}$ | $\mathbf{16.44 \pm 0.01}$ |
| ChainedGP | $6.75 \pm 0.25$ | $10.56 \pm 1.03$ | $17.31 \pm 1.06$ |

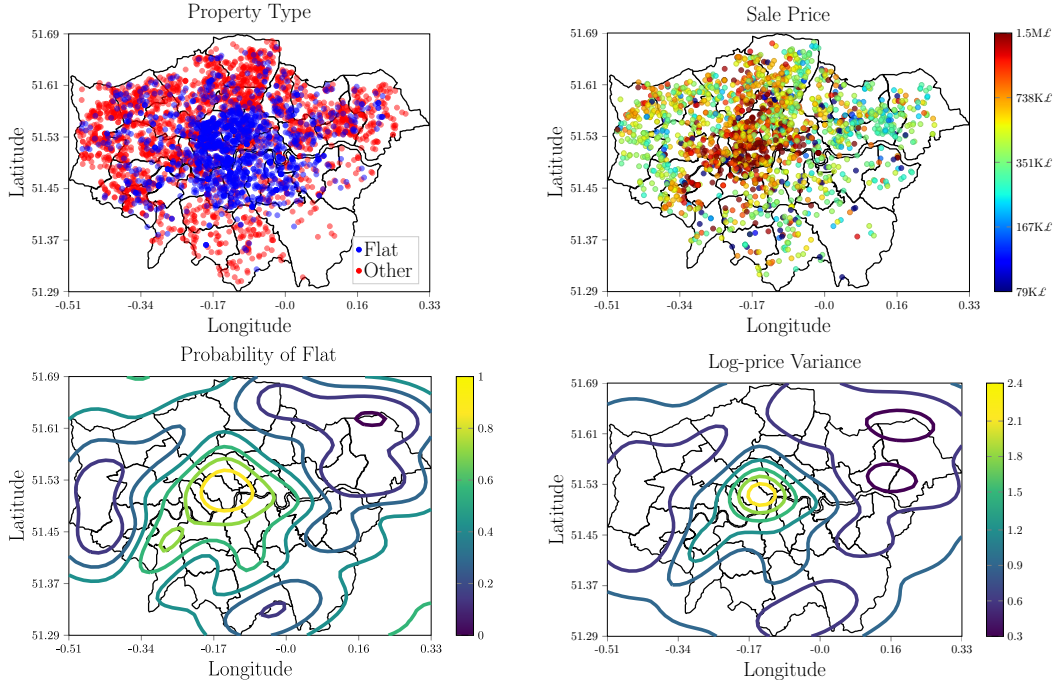

Figure 3: Results for spatial modeling of heterogeneous data. (Top row) $10\%$ of training samples for the Greater London County. Binary outputs are the type of property sold in 2017 and real ones are prices included in sale contracts. (Bottom row) Test prediction curves for $N_{\text{test}} = 2,500$ inputs.

Results in Figure (3) show a portion of the entire heterogeneous dataset and its test prediction curves. We obtained a global NLPD score of $16.44 \pm 0.01$ using the MOGP and $17.31 \pm 1.06$ in the independent outputs setting (both $\times 10^{-2}$). There is an improvement in performance when training our multi-output model even in large scale datasets. See Table (2) for scores per each output.

**High Dimensional Input Data:** In our last experiment, we tested our MOGP model for the *arrhythmia* dataset in the UCI repository (`http://archive.ics.uci.edu/ml/`). We use a dataset of dimensionality $p = 255$ and $452$ samples that we divide in training, validation and test sets

(more details are in the appendix). We use our model for predicting a binary output (gender) and a continuous output (logarithmic age) and we compared against independent Chained GPs per output. The binary output is modelled as a Bernoulli distribution and the continuous one as a Gaussian. We obtained an average NLPD value of $0.0191$ for both multi-output and independent output models with a slight difference in the standard deviation.

## 5 Conclusions

In this paper we have introduced a novel extension of multi-output Gaussian Processes for handling heterogeneous observations. Our model is able to work on large scale datasets by using sparse approximations within stochastic variational inference. Experimental results show relevant improvements with respect to independent learning of heterogeneous data in different scenarios. In future work it would be interesting to employ convolutional processes (CPs) as an alternative to build the multi-output GP prior. Also, instead of typing hand-made definitions of heterogeneous likelihoods, we may consider to automatically discover them (Valera and Ghahramani, 2017) as an input block in a pipeline setup of our tool.

**Acknowledgments**

The authors want to thank Wil Ward for his constructive comments and Juan José Giraldo for his useful advice about SVI experiments and simulations. We also thank Alan Saul and David Ramírez for their recommendations about scalable inference and feedback on the equations. We are grateful to Eero Siivola and Marcelo Hartmann for sharing their Python module for heterogeneous likelihoods and to Francisco J. R. Ruiz for his illuminating help about the stochastic version of the VEM algorithm. Also, we would like to thank Juan José Campaña for his assistance on the London House Price dataset. Pablo Moreno-Muñoz acknowledges the support of his doctoral FPI grant BES2016-077626 and was also supported by Ministerio de Economía of Spain under the project Macro-ADOBE (TEC2015-67719-P), Antonio Artés-Rodríguez acknowledges the support of projects ADVENTURE (TEC2015-69868-C2-1-R), AID (TEC2014-62194-EXP) and CASI-CAM-CM (S2013/ICE-2845). Mauricio A. Álvarez has been partially financed by the Engineering and Physical Research Council (EPSRC) Research Projects EP/N014162/1 and EP/R034303/1.

## Footnotes

[1]The code is publicly available in the repository `github.com/pmorenoz/HetMOGP/`

[2]This smartphone application can be found at `https://www.eb2.tech/`.

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
