[Supplementary Material]

# Heterogeneous Multi-output Gaussian Process Prediction
## Supplementary Material

**Pablo Moreno-Muñoz**[1]     **Antonio Artés-Rodríguez**[1]     **Mauricio A. Álvarez**[2]

[1]Dept. of Signal Theory and Communications, Universidad Carlos III de Madrid, Spain
[2]Dept. of Computer Science, University of Sheffield, UK
{pmoreno,antonio}@tsc.uc3m.es, mauricio.alvarez@sheffield.ac.uk

## 1   Complete derivation of heterogeneous multi-output lower bounds

$$\log p(\mathbf{y}) = \log \int p(\mathbf{y}|\mathbf{f})p(\mathbf{f}|\mathbf{u})p(\mathbf{u})d\mathbf{f}d\mathbf{u} = \log \int q(\mathbf{f},\mathbf{u})\frac{p(\mathbf{y}|\mathbf{f})p(\mathbf{f}|\mathbf{u})p(\mathbf{u})}{q(\mathbf{f},\mathbf{u})}d\mathbf{f}d\mathbf{u} \tag{1}$$

$$\mathcal{L} = \int q(\mathbf{f},\mathbf{u})\log\frac{p(\mathbf{y}|\mathbf{f})p(\mathbf{f}|\mathbf{u})p(\mathbf{u})}{q(\mathbf{f},\mathbf{u})}d\mathbf{f}d\mathbf{u} \tag{2}$$

$$= \int \prod_{d=1}^{D}\prod_{j=1}^{J_d}p(\mathbf{f}_{d,j}|\mathbf{u})q(\mathbf{u})\log\frac{p(\mathbf{y}|\mathbf{f})\prod_{d=1}^{D}\prod_{j=1}^{J_d}p(\mathbf{f}_{d,j}|\mathbf{u})p(\mathbf{u})}{\prod_{d=1}^{D}\prod_{j=1}^{J_d}p(\mathbf{f}_{d,j}|\mathbf{u})q(\mathbf{u})}d\mathbf{f}d\mathbf{u} \tag{3}$$

$$= \int \prod_{d=1}^{D}\prod_{j=1}^{J_d}p(\mathbf{f}_{d,j}|\mathbf{u})q(\mathbf{u})\log p(\mathbf{y}|\mathbf{f})d\mathbf{f}d\mathbf{u} - \sum_{q=1}^{Q}\text{KL}\big(q(\mathbf{u}_q)||p(\mathbf{u}_q)\big) \tag{4}$$

$$= \int \prod_{d=1}^{D}\prod_{j=1}^{J_d}\int p(\mathbf{f}_{d,j}|\mathbf{u})q(\mathbf{u})d\mathbf{u}\log p(\mathbf{y}|\mathbf{f})d\mathbf{f} - \sum_{q=1}^{Q}\text{KL}\big(q(\mathbf{u}_q)||p(\mathbf{u}_q)\big) \tag{5}$$

$$= \int q(\mathbf{f})\log p(\mathbf{y}|\mathbf{f})d\mathbf{f} - \sum_{q=1}^{Q}\text{KL}\big(q(\mathbf{u}_q)||p(\mathbf{u}_q)\big) \tag{6}$$

$$= \int q(\mathbf{f})\sum_{d=1}^{D}\log p(\mathbf{y}_d|\widetilde{\mathbf{f}}_d)d\mathbf{f} - \sum_{q=1}^{Q}\text{KL}\big(q(\mathbf{u}_q)||p(\mathbf{u}_q)\big) \tag{7}$$

$$= \sum_{d=1}^{D}\int q(\widetilde{\mathbf{f}}_d)\log p(\mathbf{y}_d|\widetilde{\mathbf{f}}_d)d\widetilde{\mathbf{f}}_d - \sum_{q=1}^{Q}\text{KL}\big(q(\mathbf{u}_q)||p(\mathbf{u}_q)\big) \tag{8}$$

$$= \sum_{d=1}^{D}\mathbb{E}_{q(\widetilde{\mathbf{f}}_d)}\big[\log p(\mathbf{y}_d|\widetilde{\mathbf{f}}_d)\big] - \sum_{q=1}^{Q}\text{KL}\big(q(\mathbf{u}_q)||p(\mathbf{u}_q)\big). \tag{9}$$

## 2   Variational Expectations

To compute the following integrals

$$\mathbb{E}_{q(\widetilde{\mathbf{f}}_d)}\big[\log p(\mathbf{y}_d|\widetilde{\mathbf{f}}_d)\big] = \int q(\widetilde{\mathbf{f}}_d)\log p(\mathbf{y}_d|\widetilde{\mathbf{f}}_d)d\widetilde{\mathbf{f}}_d, \tag{10}$$

we make use of Gaussian-Hermite quadratures. So, considering an univariate case for simplicity, expectations can be approximated as

$$\mathbb{E}_{q(\mathbf{f}_{d,1})}\big[\log p(\mathbf{y}_d|\mathbf{f}_{d,1})\big] \approx \frac{1}{\sqrt{\pi}}\sum_{s=1}^{S}w_s\log p(\mathbf{y}_d|\sqrt{2\mathbf{v}_{d,1}}\mathbf{f}_s + \mathbf{m}_{d,1}), \tag{11}$$

where $\mathbf{m}_{d,1}$ and $\mathbf{v}_{d,1}$ are the mean and variance of the variational distribution $q(\mathbf{f}_{d,1})$. In addition, the pair of values $w_s, \mathbf{f}_s$ is obtained by taking a chosen number $S$ of points from the Hermite polynomial $H_n(x) = (-1)^n e^{x^2}\frac{d^n}{dx^n}e^{-x^2}$. Note that this process must be done sequentially for multivariate expectations, which results in a multidimensional sum with an storage cost of $\mathcal{O}(S^{D_t})$ where $D_t$ is the number of output functions involved in the integral.

# 3 Predictive mean and variance of output $\mathbf{y}_d^*$

$$\mathbb{E}\big[\mathbf{y}_d^*|\mathbf{y}\big] = \int \mathbb{E}\big[\mathbf{y}_d^*|\widetilde{\mathbf{f}}_d^*\big]q(\widetilde{\mathbf{f}}_d^*)d\widetilde{\mathbf{f}}_d^* \tag{12}$$

$$\text{var}\big[\mathbf{y}_d^*|\mathbf{y}\big] = \int \text{var}\big[\mathbf{y}_d^*|\widetilde{\mathbf{f}}_d^*\big]q(\widetilde{\mathbf{f}}_d^*)d\widetilde{\mathbf{f}}_d^* + \int \mathbb{E}^2\big[\mathbf{y}_d^*|\widetilde{\mathbf{f}}_d^*\big]q(\widetilde{\mathbf{f}}_d^*)d\widetilde{\mathbf{f}}_d^* - \mathbb{E}^2\big[\mathbf{y}_d^*|\mathbf{y}\big]^2 \tag{13}$$

# 4 Gradients w.r.t. $q(\mathbf{u})$

First, the bound derivatives w.r.t. $\boldsymbol{\mu}_{\mathbf{u}_q}$ are

$$\frac{\partial}{\partial\boldsymbol{\mu}_{\mathbf{u}_q}}\mathcal{L} = \sum_{d=1}^{D}\underbrace{\frac{\partial}{\partial\boldsymbol{\mu}_{\mathbf{u}_q}}\mathbb{E}_{q(\widetilde{\mathbf{f}}_d)}\big[\log p(\mathbf{y}_d|\widetilde{\mathbf{f}}_d)\}}_{\text{VE part}} - \underbrace{\frac{\partial}{\partial\boldsymbol{\mu}_{\mathbf{u}_q}}\text{KL}\big(q(\mathbf{u}_q)||p(\mathbf{u}_q)\big)}_{\text{KL part}}, \tag{14}$$

where the KL part w.r.t. $\boldsymbol{\mu}_{\mathbf{u}_q}$ is

$$\frac{\partial}{\partial\boldsymbol{\mu}_{\mathbf{u}_q}}\text{KL}\big(q(\mathbf{u}_q)|p(\mathbf{u}_q)\big) = \mathbf{K}_q^{-1}\boldsymbol{\mu}_{\mathbf{u}_q}, \tag{15}$$

and the VE part w.r.t. $\boldsymbol{\mu}_{\mathbf{u}_q}$ yields

$$\frac{\partial}{\partial\boldsymbol{\mu}_{\mathbf{u}_q}}\mathbb{E}_{q(\widetilde{\mathbf{f}}_d)}\big[\log p(\mathbf{y}_d|\widetilde{\mathbf{f}}_d))\big] = \mathbb{E}_{q(\mathbf{F}^t)}\big[\underbrace{\frac{\partial}{\partial\widetilde{\mathbf{f}}_d}\log p(\mathbf{y}_d|\widetilde{\mathbf{f}}_d)}_{\text{See Likelihoods}}\big]\frac{\partial\widetilde{\mathbf{m}}_d}{\partial\boldsymbol{\mu}_{\mathbf{u}_q}}. \tag{16}$$

Secondly, the bound derivatives w.r.t. $\mathbf{S}_{\mathbf{u}_q}$ are

$$\frac{\partial}{\partial\mathbf{S}_{\mathbf{u}_q}}\mathcal{L} = \sum_{d=1}^{D}\underbrace{\frac{\partial}{\partial\mathbf{S}_{\mathbf{u}_q}}\mathbb{E}_{q(\widetilde{\mathbf{f}}_d)}\big[\log p(\mathbf{y}_d|\widetilde{\mathbf{f}}_d)\big]}_{\text{VE part}} - \underbrace{\frac{\partial}{\partial\mathbf{S}_{\mathbf{u}_q}}\text{KL}\big(q(\mathbf{u}_q)||p(\mathbf{u}_q)\big)}_{\text{KL part}}, \tag{17}$$

where the KL part w.r.t. $\mathbf{S}_{\mathbf{u}_q}$ is

$$\frac{\partial}{\partial\mathbf{S}_{\mathbf{u}_q}}\text{KL}\big(q(\mathbf{u}_q)||p(\mathbf{u}_q)\big) = \mathbf{K}_q^{-1} - \frac{1}{2}\text{diag}(\mathbf{K}_q^{-1}) - \frac{1}{2}\mathbf{S}_{\mathbf{u}_q}^{-1}, \tag{18}$$

and the VE part w.r.t. $\mathbf{S}_{\mathbf{u}_q}$ yields

$$\frac{\partial}{\partial\mathbf{S}_{\mathbf{u}_q}}\mathbb{E}_{q(\widetilde{\mathbf{f}}_d)}\big[\log p(\mathbf{y}_d|\widetilde{\mathbf{f}}_d)\big] = \frac{1}{2}\mathbb{E}_{q(\widetilde{\mathbf{f}}_d)}\big[\underbrace{\frac{\partial^2}{\partial\widetilde{\mathbf{f}}_d^2}\log p(\mathbf{y}_d|\widetilde{\mathbf{f}}_d)}_{\text{See Likelihoods}}\big]\frac{\partial\widetilde{\mathbf{v}}_d}{\partial\mathbf{S}_{\mathbf{u}_q}}. \tag{19}$$

where $\widetilde{\mathbf{m}}_d$ and $\widetilde{\mathbf{v}}_d$ are the corresponding mean and variance of the variational distribution $q(\widetilde{\mathbf{f}}_d)$. Each one of the variational expectations on the functional derivatives is different for a given heterogeneous

likelihood (see below). The gradients identities in (16) and (19) are similar to the ones used in Opper and Archambeau (2009); Hensman et al. (2013); Saul et al. (2016). This means to use

$$\frac{\partial}{\partial \sigma} \mathbb{E}_{\mathcal{N}(x|\mu,\sigma^2)}\big[f(x)\big] \;=\; \mathbb{E}_{\mathcal{N}(x|\mu,\sigma^2)}\big[\frac{\partial}{\partial x} f(x)\big], \tag{20}$$

$$\frac{\partial}{\partial \mu} \mathbb{E}_{\mathcal{N}(x|\mu,\sigma^2)}\big[f(x)\big] \;=\; \frac{1}{2}\mathbb{E}_{\mathcal{N}(x|\mu,\sigma^2)}\big[\frac{\partial^2}{\partial x^2} f(x)\big]. \tag{21}$$

# 5 Gradients w.r.t hyperparameters

Applying the *chain-rule* and assuming the matrix derivatives $\frac{\partial \mathbf{A}}{\partial \boldsymbol{\theta}}$ and $\frac{\partial \mathbf{A}}{\partial \mathbf{Z}}$ given for any arbitrary matrix $\mathbf{A}$ dependent on the hyperparameters, we must compute the following gradients:

$$\frac{\partial \mathcal{L}}{\partial \mathbf{K}_q}, \frac{\partial \mathcal{L}}{\partial \mathbf{K}_{\mathbf{f}_{d,j}\mathbf{u}_q}} \text{ and } \frac{\partial \mathcal{L}}{\partial \text{diag}(\mathbf{K}_{\mathbf{f}_{d,j}\mathbf{f}_{d,j}})}. \tag{22}$$

In this section, we denote $\mathbf{K}_{dq} = \mathbf{K}_{\mathbf{f}_{d,j}\mathbf{u}_q}$ and $\mathbf{K}_{\text{diag}} = \text{diag}(\mathbf{K}_{\mathbf{f}_{d,j}\mathbf{f}_{d,j}})$ for simplicity in the following expressions. Then,

$$\frac{\partial \mathcal{L}}{\partial \mathbf{K}_q} \;=\; \sum_{d=1}^{D} \frac{\partial}{\partial \mathbf{K}_q} \mathbb{E}_{q(\widetilde{\mathbf{f}}_d)}\big[\log p(\mathbf{y}_d|\widetilde{\mathbf{f}}_d)\big] - \sum_{q=1}^{Q} \frac{\partial}{\partial \mathbf{K}_q} \text{KL}\big(q(\mathbf{u}_q)||p(\mathbf{u}_q)\big), \tag{23}$$

$$\frac{\partial \mathcal{L}}{\partial \mathbf{K}_{dq}} \;=\; \sum_{d=1}^{D} \sum_{j=1}^{J_d} \frac{\partial}{\partial \mathbf{K}_{dq}} \mathbb{E}_{q(\widetilde{\mathbf{f}}_d)}\big[\log p(\mathbf{y}_d|\widetilde{\mathbf{f}}_d)\big], \tag{24}$$

$$\frac{\partial \mathcal{L}}{\partial \mathbf{K}_{\text{diag}}} \;=\; \sum_{d=1}^{D} \sum_{j=1}^{J_d} \frac{\partial}{\partial \mathbf{K}_{\text{diag}}} \mathbb{E}_{q(\widetilde{\mathbf{f}}_d)}\big[\log p(\mathbf{y}_d|\widetilde{\mathbf{f}}_d)\big]; \tag{25}$$

where

$$\frac{\partial}{\partial \mathbf{K}_q} \mathbb{KL}\big(q(\mathbf{u}_q)||p(\mathbf{u}_q)\big) = \frac{1}{2}\left( -(\mathbf{K}_q^{-1}\mathbf{S}_{\mathbf{u}_q}\mathbf{K}_q^{-1})^\top - (\mathbf{K}_q^{-1})^\top \boldsymbol{\mu}_{\mathbf{u}_q}\boldsymbol{\mu}_{\mathbf{u}_q}^\top (\mathbf{K}_q^{-1})^\top + (\mathbf{K}_q^{-1})^\top \right), \tag{26}$$

and

$$\frac{\partial}{\partial \mathbf{K}_q} \mathbb{E}_{q(\widetilde{\mathbf{f}}_d)}\big[\log p(\mathbf{y}_d|\widetilde{\mathbf{f}}_d)\big] \;=\; \frac{\partial}{\partial \widetilde{\mathbf{m}}_d} \mathbb{E}_{q(\widetilde{\mathbf{f}}_d)}\big[\log p(\mathbf{y}_d|\widetilde{\mathbf{f}}_d)\big] \frac{\partial \widetilde{\mathbf{m}}_d}{\partial \mathbf{K}_q} \tag{27}$$

$$+ \frac{\partial}{\partial \widetilde{\mathbf{v}}_d} \mathbb{E}_{q(\widetilde{\mathbf{f}}_d)}\big[\log p(\mathbf{y}_d|\widetilde{\mathbf{f}}_d)\big] \frac{\partial \widetilde{\mathbf{v}}_d}{\partial \mathbf{K}_q},$$

$$\frac{\partial}{\partial \mathbf{K}_{dq}} \mathbb{E}_{q(\widetilde{\mathbf{f}}_d)}\big[\log p(\mathbf{y}_d|\widetilde{\mathbf{f}}_d)\big] \;=\; \frac{\partial}{\partial \widetilde{\mathbf{m}}_d} \mathbb{E}_{q(\widetilde{\mathbf{f}}_d)}\big[\log p(\mathbf{y}_d|\widetilde{\mathbf{f}}_d)\big] \frac{\partial \widetilde{\mathbf{m}}_d}{\partial \mathbf{K}_{dq}} \tag{28}$$

$$+ \frac{\partial}{\partial \widetilde{\mathbf{v}}_d} \mathbb{E}_{q(\widetilde{\mathbf{f}}_d)}\big[\log p(\mathbf{y}_d|\widetilde{\mathbf{f}}_d)\big] \frac{\partial \widetilde{\mathbf{v}}_d}{\partial \mathbf{K}_{dq}},$$

$$\frac{\partial}{\partial \mathbf{K}_{\text{diag}}} \mathbb{E}_{q(\widetilde{\mathbf{f}}_d)}\big[\log p(\mathbf{y}_d|\widetilde{\mathbf{f}}_d)\big] \;=\; \frac{\partial}{\partial \widetilde{\mathbf{v}}_d} \mathbb{E}_{q(\widetilde{\mathbf{f}}_d)}\big[\log p(\mathbf{y}_d|\widetilde{\mathbf{f}}_d)\big] \frac{\partial \widetilde{\mathbf{v}}_d}{\partial \mathbf{K}_{\text{diag}}}.$$

# 6 Likelihoods and link functions

To include any new distribution, we must derive the following expressions for each heterogeneous likelihood $p(\mathbf{y}_d|\widetilde{\mathbf{f}}_d)$:

1. Log-Likelihood function $\log p(\mathbf{y}_d|\widetilde{\mathbf{f}}_d)$ for VE and the predictive distribution.

2. First order derivatives $\frac{\partial}{\partial \widetilde{\mathbf{f}}_d} \log p(\mathbf{y}_d|\widetilde{\mathbf{f}}_d)$ for VE in gradients.

3. Second order derivatives $\frac{\partial^2}{\partial \widetilde{\mathbf{f}}_d^2} \log p(\mathbf{y}_d|\widetilde{\mathbf{f}}_d)$ for VE in gradients.

4. Mean $\mathbb{E}\big[\mathbf{y}_d|\widetilde{\mathbf{f}}_d\big]$ and variance $\text{var}\big[\mathbf{y}_d|\widetilde{\mathbf{f}}_d\big]$ for predictive point-estimates.

Table 1: Used link transformations between latent parameter functions (LPFs) $\mathbf{f}$ and heterogeneous likelihoods. Note that many other valid mappings between parameters and LPFs are allowed.

| Likelihood | Linked Parameters | Number of LPFs $\mathbf{f}$ |
|---|---|---|
| Gaussian | $\mu(\mathbf{x}) = \mathbf{f}, \sigma(\mathbf{x})$ | 1 |
| Heteroscedastic Gaussian | $\mu(\mathbf{x}) = \mathbf{f}_1, \sigma(\mathbf{x}) = \exp(\mathbf{f}_2)$ | 2 |
| Bernoulli | $\rho(\mathbf{x}) = \frac{\exp(\mathbf{f})}{1+\exp(\mathbf{f})}$ | 1 |
| Categorical | $\rho_k(\mathbf{x}) = \frac{\exp(\mathbf{f}_k)}{1+\sum_{k'=1}^{K-1}\exp(\mathbf{f}_{k'})}$ | K-1 |
| Exponential | $\beta(\mathbf{x}) = \exp(-\mathbf{f})$ | 1 |
| Poisson | $\lambda(\mathbf{x}) = \exp(\mathbf{f})$ | 1 |
| Gamma | $a(\mathbf{x}) = \exp(\mathbf{f}_1), b(\mathbf{x}) = \exp(\mathbf{f}_2)$ | 2 |
| Beta | $a(\mathbf{x}) = \exp(\mathbf{f}_1), b(\mathbf{x}) = \exp(\mathbf{f}_2)$ | 2 |

# 7 Experiments and hyperparameter setup

The code for experiments is written in Python and publicly available. It can be found in the repository `https://github.com/pmorenoz/HetMOGP/`, where we have made use of the GPy software library, specially indicated for Gaussian processes simulations. When the method was implemented on its non-scaled version (order of magnitude in the observation set around $10^2$-$10^3$ samples), we used the LBFGS-B algorithm implemented in the *paramz* module. For higher dimensions (number of input samples($> 10^3$)), we loaded the stochastic version of the model. The general SVI setting was a *batch* size of 500 samples in every output and the step rate was set to 0.05. The optimization algorithm for SVI is ADADELTA included in the *climin* library and we set the *momentum* parameter to 0.9.

We compared our MOGP method with an independent output fitting for every likelihood. During the experiments, we refer to the Chained GP model from Saul et al. (2016). We adapted this model to work with a greater number of likelihoods than the original ones presented in its original version.

**Initialization of hyperparameters:** We initialize hyperparameters $\mathbf{Z}$, $\{\mathbf{B}_q\}_{q=1}^{Q}$ as follows: For the inducing points $\mathbf{Z}$ we start by taking equally separated points per each input dimension in $\mathbf{X}$. Secondly, linear operators of the MOGP prior were initialized randomly using a Gaussian $\mathcal{N}(\mu = 1, \sigma = 1)$. Hyperparameters of all function covariances $k_q(\cdot, \cdot)$ with an RBF kernel were initialized with lengthscales set to 0.05 in order to begin with sufficient flexibility. On the other hand, amplitude variances were set to initial values of 0.5.

**Initialization of variational parameters:** We use a general initialization for all variational distributions $q(\mathbf{u})$. Mean parameters $\boldsymbol{\mu}_{\mathbf{u}_q}$ are sampled from a Gaussian distribution $\mathcal{N}(\mu, \sigma = 2)$ where $\mu$ is similarly obtained from $\mathcal{N}(1, 1)$. For the covariance matrices $\mathbf{S}_{\mathbf{u}_q}$ we set it to the identity matrix $\mathbf{I}$.

**Stochastic variational EM algorithm:** In the non-stochastic scenario, the variational EM algorithm (VEM) was programmed to switch between fixed and unfixed variational parameters or hyperparameters sequentially. For SVI, we adapted the code to load or unload gradients depending of the VEM step. When modeling was too sensitive to hyperparameters changes, we runned three VE steps of the stochastic noisy gradient for every one of the VM. This was the case for the London dataset.

**Heterogeneous likelihood syntaxes:** In our code, we implemented a simple manner to define the heterogeneous likelihood for combinations of an arbitrary number of likelihood functions. The assignment of LPFs to parameters is done automatically. Some examples are given below:

```
> likelihood_list = [Gaussian(), Gaussian(sigma=0.5), Exponential()]

> likelihood_list = [HetGaussian(), Bernoulli(), Categorical(K=3)]

> likelihood_list = [Gamma(), Categorical(K=5)]
```

**Additional experiments:** We have also evaluated our heterogeneous model on new experiments and different versions of the previously presented ones. These additional results are: [Behavior] Heteroscedastic Gaussian (distance from Home) and Poisson (number of active *apps*) outputs. We removed the Bernoulli output (presence/absence), [London] We replaced property type by type of contract (freehold or leasehold) as the new binary output. [Arrhythmia] High-dimensional input, Categorical output with $K = 3$: normal, arrhythmia or other. [Synthetic] Arbitrary combination of 10 outputs (Gaussian, Bernoulli and Exponential). The Table below shows test-NLPD results for every dataset. Notice that the proposed model outperforms the Chained GP for all the suggested experiments.

Table 2: Test-NLPD ($\times 10^{-2}$) results for all additional experiments.

| Experiment | HetMOGP | ChainedGP |
|---|---|---|
| Behavior | **0.1883 ± 0.0027** | 0.3281 ± 0.0122 |
| London | **0.1646 ± 0.0153** | 0.2144 ± 0.0402 |
| Arrhythmia | **0.0987 ± 0.0004** | NA |
| Synthetic⋆ | **1.7371 ± 0.4128** | 4.3182 ± 2.4402 |

Figure 1: Results for spatial modeling of heterogeneous data. (Top row) $10\%$ of training samples for the Greater London County. Binary outputs are the type of contract (freehold or leasehold) in 2017 and real ones are the prices included in sale contracts. (Bottom row) Test prediction curves for $N_{\text{test}} = 2,500$ inputs.