[Reviews · NeurIPS 2018]

Reviewer 1



*** Update after author feedback *** Thank you for your feedback. I didn't understand first that the model cannot only be used for the LMC models, but also for convolutions. This is great and extends the usefulness of the model further. ***************************************** The paper introduces a new multi-output model for heterogenous outputs, i.e. each output can have its own likelihood function. The models builds upon the linear model of coregionalization for coupling the latent regressors of the multiple outputs and uses stochastic variational inference to allow for non-Gaussian likelihoods. The main concern I have about the paper is its novelty in terms of technical contribution. The paper combines the LMC model for the Gaussian case (Bonilla et al., 2014, I missed the citation!) with progress for other types of likelihoods (Saul et al, 2016). Are both models special cases of the aforementioned model or are there any differences in the inference scheme? However, the paper is well written and the experiments are convincing which is why I would recommend a "weak accept". The paper also comes along with a Python implementation that allows to quickly try out multi-output models with new data types which could lead to a wider spread of multi-output models for that type of data. Minors: L 186: Why would the Monte Carlo approach be slower than Gaussian-Hermite quadrature? Please be more explicit. L 291: Please state the standard deviations. L 267: What are the features? I first thought it is the time but then the sentence (270-272) does not make any sense. L 268: The dataset consists of 604,800 samples but only 750 samples are used for training. How is the performance w.r.t. runtime/NLPD if the training setsize is increased? Literature: - Nguyen, Trung V., and Edwin V. Bonilla. "Collaborative Multi-output Gaussian Processes." UAI. 2014. - Saul, Alan D., et al. "Chained gaussian processes." Artificial Intelligence and Statistics. 2016.

Reviewer 2



The paper proposes using different likelihoods for each of the outputs of a multi-output Gaussian process. The paper addresses most areas: a model, an approximation (with variational bounds), stochastic inference for large data sets, hyper-parameter tuning, how prediction is done in the approximate model, experiments on synthetic data, and experiments on real-world data. Hence, the paper is very thorough. However, there a a number of shortcomings which will be highlighted below. While I have not seen any paper prior to this that explicitly places different likelihoods on the outputs of a multi-output Gaussian process, it is not hard imagine this --- especially when variational inference is used which reduces the (log-)likelihood terms to a summation. The difficulty is to find convincing applications/data for which such a model is useful. I think the Human Behaviour Data is a good fit for this model, except that the presence/absence-at-home output is somewhat redundant and contrived --- isn't this output a threshold version of the distance-from-home output? I also find the choice of the London House Price Data rather inappropriate --- the authors have used the property-type as an output when it is best used as an input. For the High Dimensional Input Data, the authors have chosen to predict the gender and age, which are attributes/covariates/inputs in the data set, while side-stepping the more interesting and important and original intended task of distinguish between "the presence and absence of cardiac arrhythmia and to classify it in one of the 16 groups." Instead of these last two data sets, I recommend that the authors concentrate on the Human Behavior Data --- for example, using a counts model for the number of Whatsapp messages sent; and/or analyzing the effect of the present/absence-at-home output. In addition, I also think a covariance function that incorporate periodicity is a better fit than a vanilla RBF kernel for this data. L65: A better characterization of the method may be to say that it *combines* MOGP with Chained GP, and that it *generalises* L154 and L157: Here, the concepts of independence in the model and independence in the approximation are not clearly stated. In section 4, it may be clearer to state that Chained GP with Bernoulli is simply the classical binary-classification Gaussian processes. Minor: The two [Valera el al. 2017] references need to be given different labels. [Quality] This submission is technically sound. However, I feel that the experiments are lacking to fully evaluation the benefits of this model. In addition, the authors should compare with the work of Yang et al. 2009 using the data sets therein (both simulated and the asthma data); or the work of Valera et al. 2017 using the data sets therein. [Clarity] This submission can be clearer by addressing some of the points above. [Originality] This paper is a new but rather obvious combination of previous work. [Significance] I think the work, especially with the released code, will be widely used in data sets of this nature. [Comments after author rebuttal] The reply has addressed my reservations on "convincing applications and data", and I have revised my score upwards.

Reviewer 3



## [Updated after author feedback] Thank you for your feedback. I am happy to see the updated results and I hope you will add them to the paper. While I agree with the other reviewers that the individual parts of the idea are not new, I find the combination elegant - a whole that is greater than the sum of its parts. I will, therefore, keep my score. ## Summary The paper presents an extension to multi-output Gaussian processes enabling them to deal with heterogeneous outputs specified by different distributions, thus requiring different likelihood functions. By assuming that each output is completely specified by a distribution, the task is to infer the parameters of the distributions. Each parameter is modelled as the (non-linear transformation of the) output of a latent parameter function f, which itself is a linear combination of Q latent functions u. The main novelty is then to impose a multi-output GP prior on f, allowing the model to learn correlations between all parameters for all outputs. The authors introduce inducing variables and derive bounds allowing for stochastic variational inference, thus making the model applicable to large datasets. ## Quality The paper is of high technical quality. Some of the derivations are left out or moved to the supplementary material. Is some sense this is justified, as they follow previous work which is adequately cited. However, a pointer to the gradient specification in the appendix would be appreciated. The authors present four experiments for empirical analysis. I hate to be that reviewer asking for additional experiments, but I think it would be interesting to see how the model performs on a dataset with a large number of outputs. The maximum number of outputs evaluated is three, whereas a high-dimensional input problem is considered. A high-dimensional output problem is, in my opinion, at least as interesting. Using synthetic data as in the first problem would be just fine. In table 1 and 2, I cannot find information on how the uncertainties were obtained. That should be included. Also, I am unsure what is meant by the "Global" column. How does the covariance function look here? ## Clarity The paper is clear and well-written. The authors have clearly spent time on the writing and the structure. The main problem and contribution are both clearly stated. I really like the paragraph headlines in section 2.2 - they provide a structured and easy to follow overview. One suggestion is to consider making a sketch of the setup. With both latent functions and parameter functions, things quickly get complex with lots of subscripts and indices. Not that the text is unclear, it is just a complex problem to wrap your head around. ## Originality Related work section is clear and concise. I could not find papers that have been missed. To my knowledge, a multi-output GP model capable (in principle) of handling any type and number of outputs has not been proposed before. ## Significance The paper addresses an important issue, namely learning correlations between heterogeneous outputs using GPs. The authors further make it scalable to large datasets by casting it in the stochastic variational inference framework. This method is an important contribution to the field of GPs. ## Comments line 100: "there is no always" -> "there is not always". line 170: I believe the mean should be \mu_{u_q} instead of u_q? line 291: "slighty difference" -> "slight difference"